# Emotional Eating in Relation to Worries and Psychological Distress Amid the COVID-19 Pandemic: A Population-Based Survey on Adults in Norway

**DOI:** 10.3390/ijerph18010130

**Published:** 2020-12-27

**Authors:** Mitra Bemanian, Silje Mæland, Rune Blomhoff, Åsgeir Kjetland Rabben, Erik Kristoffer Arnesen, Jens Christoffer Skogen, Lars Thore Fadnes

**Affiliations:** 1Department of Global Public Health and Primary Care, Faculty of Medicine, University of Bergen, 5020 Bergen, Norway; Silje.Maeland@uib.no (S.M.); lars.fadnes@uib.no (L.T.F.); 2Bergen Addiction Research, Department of Addiction Medicine, Haukeland University Hospital, 5021 Bergen, Norway; 3Research Unit for General Practice in Bergen, The Norwegian Research Center, NORCE, 5008 Bergen, Norway; 4Department of Nutrition, Institute of basic medical sciences, University of Oslo, 0317 Oslo, Norway; rune.blomhoff@medisin.uio.no (R.B.); erikarnesen@hotmail.com (E.K.A.); 5Department of Clinical Service, Division of Cancer Medicine, Oslo University Hospital, 0424 Oslo, Norway; 6Section for Strategy and Analysis, Bergen municipality, 5020 Bergen, Norway; Asgeir.Rabben@bergen.kommune.no; 7Department of Health Promotion, Norwegian Institute of Public Health, 5015 Bergen, Norway; Jens.Christoffer.Skogen@fhi.no; 8Department of Public Health, Faculty of Health Sciences, University of Stavanger, 4021 Stavanger, Norway; 9Alcohol & Drug Research Western Norway, Stavanger University Hospital, 4010 Stavanger, Norway

**Keywords:** mental health, feeding behavior, comfort eating, sugar-sweetened beverages, dietary sugars

## Abstract

Population-based studies have revealed a high occurrence of self-reported psychological distress symptoms during the early phases of the coronavirus disease 2019 (COVID-19) pandemic. Stress and negative affect can lead to emotional eating, which in turn can have negative outcomes on health. In this population-based study, 24,968 Norwegian inhabitants participated in an electronic questionnaire including structured questions on dietary habits, emotional eating, psychological distress symptoms, and COVID-19-related worries. The study took place during April 2020 after around six weeks of interventions to tackle the first wave of the COVID-19 pandemic. Overall, emotional eating was reported in 54% of the population and was markedly more frequent in female participants. Worries related to consequences of the pandemic were associated with increased emotional eating, and the association was stronger for worries related to personal economy—odds ratios (OR) 1.7 (95% confidence interval (CI95%) 1.5–1.9)—compared to worries related to health—OR 1.3 (CI95% 1.2–1.5). Psychological distress had a strong association with emotional eating—OR 4.2 (CI95% 3.9–4.4). Correspondingly, the intake of high-sugar foods and beverages was higher for those with substantial COVID-19-related worries and those with psychological distress compared to the overall population.

## 1. Introduction

The coronavirus disease 2019 (COVID-19) pandemic constitutes a worldwide state of emergency that is unparalleled in recent times [1]. Countries all over the world have implemented unprecedented non-pharmaceutical interventions (NPIs) in order to limit the spread of the novel viral disease, including regional- and national-level lockdowns and quarantine. As communities were closing down, concerns were raised about the impact of these interventions on other aspects of public health, including mental health [2]. In the wake of the first phase of the pandemic, evidence of its impact on mental health is emerging. Although limited, most current studies show an increased burden of mental health symptoms among psychiatric patients and healthcare workers and lower psychological well-being in the general population amid the COVID-19 outbreak [3]. Population-based studies have also revealed a high occurrence of self-reported psychological distress during the early phases of the pandemic in different populations [4,5,6].

Stress and emotional state influence the eating behavior. To many, stress and negative mood can induce loss of appetite and hypophagia [7,8]. However, for a major subset of individuals, negative emotions and stress cause them to eat more—a type of eating known as emotional eating [9]. Stress is known to induce a shift in individuals, making them favor hyperpalatable foods, i.e., calorie-dense foods high in fat and/or sugar content, and this tendency is stronger in emotional eaters [7,10]. Even under habitual circumstances, emotional eaters consume more energy-dense and sweet snack foods than others [11,12]. There is evidence emerging that points to the role of emotional eating as a behavioral mechanism mediating the association of depression with adverse weight gain [13,14]. Thus, emotional eating may pose an additional health burden to those vulnerable to eating more, and more unhealthily, during moments of hardship and distress. 

COVID-19 represents a major disruption on the day-to-day life of most people in affected areas. There is reason to believe that the pandemic has an impact on the eating habits not only due to the practical effects of the lockdown, but also through its effect on perceived stress and psychological well-being. Current studies on eating habits amid the COVID-19 outbreak reveal a shift in self-reported eating towards increased appetite and overall food consumption, as well as increased snacking in-between meals [15,16,17]. A study regarding the most critical phase of the first Italian lockdown shows a substantial prevalence of emotional eating in the population, in particular among those that reported feeling depressed or anxious [4]. Similarly, a Chinese study on emotional eating and gestational weight gain amid the pandemic found that women who worried about COVID-19 had higher emotional eating scores [18]. Our aim was to assess the prevalence of emotional eating and its association with worries and psychological distress amid the COVID-19 lockdown through a large-scale population-based study in a major city in Norway. Our main research questions were:How prevalent was emotional eating among females and males in different age groups during the COVID-19 lockdown?Did the consumption of high-sugar foods and beverages differ between the overall population and those experiencing COVID-19-related worries or psychological distress?Was emotional eating associated with levels of COVID-19-related worries or psychological distress?

## 2. Materials and Methods

### 2.1. Survey Methodology

A random selection of 81,170 individuals from a total of 224,000 adult inhabitants in Bergen, Norway, were invited to participate in a survey assessing the consequences of the COVID-19 outbreak and the NPIs that were implemented. NPIs included social distancing and quarantine, travel restrictions, closing of schools and universities, mandated use of home office, restrictions on the use of public space and closed social arenas for sports, leisure, and culture activities. Participants were drawn from a contact register through the Norwegian Digitalization Agency. The questionnaire was sent to invitees by email through the web-based questionnaire tool SurveyXact. Data were collected between 15 April and 30 April. During the study period, restrictions and policies regarding the lockdown remained mainly unaltered in Norway. In total, 29,535 (36%) individuals agreed to participate in the study, and among these, 24,968 (84%) completed questionnaire items relevant to this study.

### 2.2. Questionnaire

The questionnaire contained items collecting demographic information and self-reported weight and height and items that focused on various aspects of life and health amid the first COVID-19 lockdown. Survey items relevant to this study include those assessing eating habits and emotional eating, COVID-19-related worries and symptoms of psychological distress—all of which are included in detail in the Appendix A.

In brief, participants were asked to express their level of worry concerning health-related and economic consequences of the pandemic and the lockdown on a three-point scale ranging from not worried to substantially worried. Health-related worries included fear of transmission of COVID-19 to oneself, to closed ones, or to elderly family members. Worries related to the economy included fear of being laid off or experiencing poorer personal economy. In accordance with a previous Norwegian survey, emotional eating was assessed by asking the participants to recall the number of times, during the past week, they had been comfort eating or eating more in response to feeling down or unsatisfied [19]. Scoring 5 or higher on the item was defined as “frequent emotional eating”. Participants were also asked to recall how often, on average, they had been consuming high-sugar food items and beverages during the past 30 days. In the questionnaire, high-sugar food items were defined and exemplified as cakes, biscuits, desserts, and candies, whereas high-sugar beverages included soft drinks and soda. A self-reported consumption above three times per week was considered a “frequent intake”. Psychological distress was measured using the 10-item version of the standardized questionnaire Hopkins symptom checklist (SCL-10) assessing mental health symptoms during the past 7 days, setting the threshold of distress at a mean SCL-10 score of 1.85 [20].

### 2.3. Statistical Analyses

All statistical analyses were performed using the software Stata SE 16 (StataCorp, College Station, TX, USA). Graphic presentations were produced in Microsoft Excel 15.26 (Microsoft, Redmond, WA, USA). The Pearson’s chi square test was used in order to determine the statistical significance of contingency tables on which graphic presentations were based. An ordered logistic regression model presented with odds ratios (OR) with 95% confidence intervals was used to predict the degree of emotional eating on a scale from 1 to 7 by the variables COVID-19-related worries, work-related consequences of COVID-19 and psychological distress, and sociodemographic factors. The odds ratios in ordinal logistic models give the change in odds for a one-unit increase in the Likert scale.

Assumptions including correlation/collinearity between independent variables in the model and proportional odds were checked. The proportionality of odds assumption was assessed for all predictor variables separately. Criteria for proportional odds were met, although the assessment revealed small variations for certain predictors. For the sake of transparency, a multinomial logistic regression table is presented (Appendix A). Results were considered statistically significant at *p*-values < 0.05 for all analyses. Participants with missing answers on relevant items were excluded from all analyses. Descriptive statistics with percentages and medians (with 25–75 percentiles) are also presented.

### 2.4. Ethics

The study was approved by the Norwegian Regional Committee for Ethics in Medical Research (REK 2020/131560). All participants provided electronic informed consent before responding to the emailed survey; confidentiality and the right to withdraw from the study were assured. The study conforms with the ethical principles outlined in the Declaration of Helsinki.

## 3. Results

### 3.1. Demographic Characteristics of the Population

Of the 24,968 participants, 50% were under the age of 50, and 56% were female, as shown in Table 1. The median body mass index (BMI) was 25 (interquartile range (IQR): 23–28). In total, 45% of participants reported substantial worries concerning health-related consequences of the COVID-19 pandemic, while 17% expressed substantial worries related to outcomes on personal economy. Twenty percent of the overall population scored above the threshold of psychological distress. As to consequences of the pandemic and lockdown, 16% of the participants were or had been placed in quarantine, 37% were working from home, and 8% were temporarily laid off from work.

### 3.2. Prevalence of Emotional Eating in Females and Males in Different Age Groups

Overall, 62% of females and 43% of males reported episodes of emotional eating during the past week, as shown in Figure 1. Frequent emotional eating was reported by 16% of females and 9% of males. Emotional eating was least prevalent in the oldest age groups, χ^2^ (30, *N* = 24,968) = 1200, *p* < 0.001).

### 3.3. Consumption of High-Sugar Foods and Beverages

Twenty-two percent of those with substantial worries reported a high weekly intake of high-sugar food items, compared to 19% of those without substantial worries (Figure 2). In participants with psychological distress, the numbers were 27% and 19%, respectively. As to high-sugar beverages, 11% of worried participants had a frequent weekly intake, compared to 7% of those without substantial worries. Among the psychologically distressed participants, 15% reported a high weekly intake of high-sugar beverages, compared to 8% of the non-distressed population.

### 3.4. Emotional Eating in Relation to COVID-19-Related Worries and Psychological Distress

Female participants were more inclined than men to report emotional eating (OR 1.9 (CI95% 1.8–2.0)), as shown in Table 2. When comparing participants in the age group between 18 and 30 years, those in older age categories predominantly reported less emotional eating. Conversely, participants aged 30–39 years were more inclined to report emotional eating (OR 1.3 (CI95% 1.1–1.4)). Substantial health-related worry was weakly associated with emotional eating (OR 1.3 (CI95% 1.2–1.5)), whereas substantial worry related to personal economy had a stronger association with emotional eating (OR 1.7 (CI95% 1.5–1.9)). The strongest association was found between psychological distress and emotional eating (OR 4.2 (CI95% 3.9–4.4)).

## 4. Discussion

This large population-based study presents data on the prevalence of emotional eating and its association with worries and mental distress during the COVID-19 lockdown in Norway and is the first to explore associations between emotional eating and worries related to health and personal economy during a pandemic—findings that are in line with those presented in a study on eating habits during the COVID-19 lockdown period in Italy [4]. Due to limited evidence on the baseline levels of emotional eating in the study population, we are unable to directly compare these numbers with those from pre-COVID-19 times. However, emotional eating and associated eating behaviors are closely related to stressful life events and to perceived life stress [21,22,23]. The COVID-19 pandemic and the NPIs that were implemented during the lockdown period could represent a significant source of stress to many, and it is therefore not implausible that the prevalence of emotional eating was in fact increased during the study period compared to habitual levels. Literature from another collectively stressful event, namely, an earthquake in New Zealand, showed increased over-eating in subjects that were already prone to emotional eating and who reported high levels of stress related to the event [24].

Female participants reported more emotional eating than men, consistently with findings in other studies [12,25]. An interesting hypothesis relates this gender difference to dietary restraint. Dietary restraint refers to the cognitive effort to control food and calorie intake, as is the case with dieting. In general, women exhibit higher dietary restraint than men [26]. Moreover, experimental studies have shown that dieters eat more when exposed to stress or negative emotion than non-dieters, suggesting that dieting could in fact be a possible risk factor for emotional eating [9,27]. Interestingly, di Renzo et al. found that those who had been dieting prior to the study period were more prone to emotional eating [4].

Worries concerning the consequences of COVID-19 were weakly associated with emotional eating. This association was stronger for worries related to personal economy and job security compared to those related to health and disease transmission. Possibly, participants with health-related worries could be more conscious of their eating behavior and therefore have some inherent resistance to unhealthy and emotional eating. It is also possible that the prospect of negative outcomes on personal economy could have a stronger impact on worriers than the prospect of disease transmission and contraction, i.e., it is possible that the economic worriers were, in fact, more worried. A study comparing the mediation effect of emotional eating on the association of depression with BMI in Denmark and Spain highlighted the role of unemployment in explaining this mediation in the latter country [28]. This was seen in the context of the 2008–2014 Spanish financial crisis, leading to an upsurge in unemployment. During our study period, the COVID-19 lockdown led to a more than four-fold rise in unemployment in Norway [29]. Importantly, there was a social gradient in the loss of work related to COVID-19, where employees with the lowest income were the most at risk of losing their job. In Norway, several measures to reduce economic vulnerability were implemented early in the lockdown period [30]. Thus, it can be assumed that the associations we present could have been even stronger in the absence of such measures.

Emotional eating was strongly associated with psychological distress. This is in line with findings reported in the literature which present positive associations between depressive symptoms and emotional eating [11,12,13,14]. Similarly, the intake of high-sugar foods and beverages was markedly higher in those scoring above the threshold of psychological distress compared to the overall population. Stress is known to induce a shift in individuals to favor energy-dense foods containing high amounts of fat and/or sugar, especially among emotional eaters [7,10]. An experimental study on the mood-enhancing effect of chocolate found that chocolate, in particular, highly palatable milk chocolate, had a short-lived effect on improving negative mood, and that this effect was stronger in high-degree emotional eaters than in the other participants [31]. It is unlikely that eating certain foods has an actual role in enhancing mood and lowering stress other than short-lived reward or relief [21,32]. Still, these short-term effects, or even the belief that eating certain foods will provide comfort, may potentially support a tendency towards stress-induced (over-) eating. This habit could, on the long term, represent an added health burden in the form of adverse weight gain and obesity. With regard to the current pandemic, it is worth noting that obesity and related co-morbidities are emerging as risk factors for poor outcomes in COVID-19 patients [33].

Our study clearly demonstrates that psychological distress is associated with emotional eating and a higher consumption of high-sugar foods and beverages. The same is seen, but to a lesser extent, in those reporting substantial COVID-19-related worries. Moreover, we found that worries related to personal economy and job security had a stronger association with emotional eating compared to worries related to health and disease transmission. To our knowledge, this is the first study demonstrating this. This finding raises the question of whether those vulnerable to financial stress, e.g., those with a lower socioeconomic status, are more at risk of emotional eating during periods of economic uncertainty. In fact, emotional eating could play a role in the association of low socioeconomic status with higher BMI—a role that is dependent on emotional and psychological distress [34,35]. Future research, preferably in the form of large-scale longitudinal studies, could further elucidate this and the possible preventive health measures aimed at populations at risk of making adverse food choices during moments of hardship and distress. Interestingly, recent randomized controlled trials point to dietary improvement as a promising treatment strategy for depression [36,37]. Interventions aimed at healthy eating could have a dualistic beneficial effect on improving eating behaviors, while at the same time reducing psychological distress for those vulnerable to emotional eating.

### Strengths and limitations

Our study provides an overview of the prevalence of emotional eating and its association with worry and psychological distress on a large scale amid a worldwide state of emergency. One strength of this study is its large sample size, allowing analyses with high precision and statistical power. Moreover, the study period coincided with the most invasive implementation of NPIs in Norway yet, and this offers insight into a phase that exemplifies the impact of a large-scale pandemic.

One inherent limitation of this study is its cross-sectional design, limiting causal inferences. This also prevents us from drawing conclusions about the effect of COVID-19 and related exposures on outcomes, specifically. Another limitation of this study is that it relies on self-report and therefore is subject to recall bias and dependent on the participants’ own insights. It is also based on relatively few, but validated, questions related to eating habits and could therefore provide a limited range of detail and nuance compared to a larger questionnaire. There is also an inherent selection bias in our study due to the questionnaire being written in Norwegian and distributed solely through digital means, which excludes people without access to the internet and with limited proficiency in the Norwegian language—e.g., elderly inhabitants and first-generation immigrants. Moreover, the senior citizens that did participate in this study are likely more self-reliant and healthier than those in the same age group that were unable to participate, providing a potentially unbalanced view of the oldest age group.

## 5. Conclusions

Our study has shown associations between psychological distress, COVID-19-related worries, and emotional eating amid the first phase of the pandemic in Norway. Emotional eating was reported by more than half of the population during the lockdown period and was especially prevalent among females, those expressing substantial worries concerning COVID-19-related consequences—and, in particular, among those experiencing psychological distress. The association of COVID-19-related worries with emotional eating was strongest for individuals concerned about potential consequences on personal economy and job security. In addition, we found that the reported intake of high-sugar foods and beverages was higher in the same groups compared to the general population. Emotional eating could pose an additional health burden to vulnerable populations in the form of poor food choices and adverse weight gain during this extraordinary global event.

## Figures and Tables

**Figure 1 ijerph-18-00130-f001:**
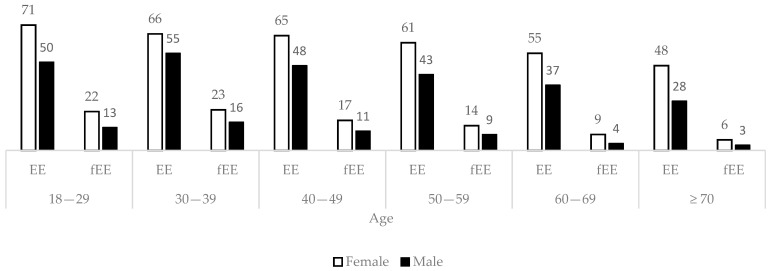
Proportions (%) of the participants reporting any degree of emotional eating (EE) and frequent emotional eating (fEE) among females (white) and males (black) in different age groups. Differences were determined to be statistically significant by means of a Pearson´s Chi Square test (*p* < 0.05). fEE = EE ≥ 5 times/week.

**Figure 2 ijerph-18-00130-f002:**
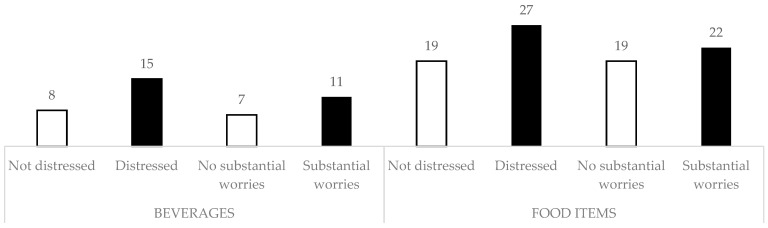
Proportions (%) of participants reporting frequent intake of high-sugar beverages (**left**) and food items (**right**), comparing those with psychological distress and substantial COVID-19-related worries with those that were not distressed and not worried. Differences were determined to be statistically significant by means of a Pearson’s Chi Square test (*p* < 0.05). Frequent consumption is defined as a self-reported intake above 3 times per week.

**Table 1 ijerph-18-00130-t001:** Demographic characteristics, worries, and mental distress in the sample population.

	Age						Total (%)
	18–29	30–39	40–49	50–59	60–69	≥70	
N (%)	3414 (14)	4228 (17)	4772 (19)	5262 (21)	4237 (17)	3055 (12)	-
Gender							
Female	2200 (64)	2530 (60)	2768 (58)	2950 (56)	2133 (50)	1363 (45)	44
Male	1214 (36)	1698 (40)	2004 (42)	2312 (44)	2104 (50)	1692 (55)	56
BMI median	23	25	25	26	26	25	25
25–75 percentiles	21–26	22–28	23–28	24–29	23–28	23–28	23–28
Education level							
Primary school	414 (13)	150 (4)	182 (4)	272 (5)	358 (9)	375 (13)	7
High school or trade school	1177 (36)	773 (19)	932 (20)	1604 (31)	1346 (32)	959 (32)	28
≤3 years of higher education	866 (26)	1046 (26)	1098 (24)	1243 (24)	938 (22)	692 (23)	24
≥4 years of higher education	828 (25)	2109 (52)	2412 (52)	2022 (39)	1531 (37)	969 (32)	40
Employment							
Employed	2178 (64)	3548 (84)	4175 (87)	4550 (86)	2415 (57)	201 (7)	68
Student/military duty	1578 (46)	251 (6)	101 (2)	31 (1)	6 (0)	3 (0)	6
Household income *							
Low	1041 (36)	522 (13)	462 (11)	364 (8)	234 (6)	314 (12)	13
Medium	1098 (38)	1944 (49)	2259 (51)	1800 (38)	1262 (35)	1204 (50)	44
High	734 (26)	1464 (37)	1678 (38)	2536 (54)	2109 (59)	886 (37)	43
Lockdown consequences							
Placed in quarantine	767 (22)	696 (16)	718 (15)	738 (14)	630 (15)	507 (17)	16
Home office	712 (21)	2069 (49)	2558 (54)	2477 (47)	1243 (29)	70 (2)	37
Temporarily laid off	505 (15)	394 (9)	358 (8)	411 (8)	216 (5)	18 (1)	8
Lost employment	76 (2)	60 (1)	38 (1)	39 (1)	23 (1)	5 (0)	1
Substantial worry							
Related to personal economy	998 (29)	1025 (24)	877 (18)	862 (16)	342 (8)	64 (2)	17
Health-related	1889 (55)	2089 (49)	2181 (46)	2507 (48)	1559 (37)	887 (29)	45
Psychological distress **	1403 (41)	1211 (29)	935 (20)	823 (16)	475 (11)	247 (8)	20

* Household income was adjusted by family size (first adult with weight 1, additional adults 0.7. and children 0.5. Low is Norwegian krone (NOK) < 250 K/year, middle is 250–500 K/year, and high is >500 K/year; these values can be converted to Euros on the basis of the currency rate of 1 May 2020 (11.422). ** Mean symptom checklist (SCL)-10 score ≥1.85. BMI, body mass index.

**Table 2 ijerph-18-00130-t002:** Ordered logistic regression of emotional eating by predictors including COVID-19-related worries and psychological distress.

	Odds Ratio	95% Confidence Interval
Gender		
Male	Reference	Reference
Female	1.9	1.8–2.0
Age		
18–29	Reference	Reference
30–39	1.3	1.1–1.4
40–49	1.1	1.0–1.2
50–59	0.93	0.85–1.0
60–69	0.75	0.69–0.82
70+	0.55	0.50–0.61
Education level		
Primary school	Reference	Reference
High or trade school	1.0	0.93–1.1
≤3 years of higher education	1.1	1.0–1.2
≥4 years of higher education	1.0	0.94–1.1
Quarantined		
No	Reference	Reference
Yes	1.2	1.1–1.3
Temporarily laid-off		
No	Reference	Reference
Yes	1.0	0.88–1.1
Health-related worries		
None	Reference	Reference
Some	1.3	1.2–1.3
Substantial	1.3	1.2–1.5
Economy-related worries		
None	Reference	Reference
Some	1.3	1.2–1.4
Substantial	1.7	1.5–1.9
Psychological distress		
SCL10 ≤ 1.85	Reference	Reference
SCL10 ≥ 1.85	4.2	3.9–4.4

## Data Availability

The data presented in this study are available on request from the corresponding author. The data are not publicly available due to privacy restrictions.

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
