# Peer review of "Emotional Eating in Relation to Worries and Psychological Distress Amid the COVID-19 Pandemic: A Population-Based Survey on Adults in Norway"

_ijerph, 2020, doi:10.3390/ijerph18010130_

Round 1

Reviewer 1 Report

Line 139-145: Please check carefully presented values (percentages) in the text (all section 3.1.) with values presented in the Table 1. They are different! Example: ”Among the 24,968 participants 50% were under the age of 50, and 56% were female…” in my opinion 60% were female (Table 1), …37% were working from home and 8 % were temporarily laid-off from work (text) and Table 1: 29 and 10%, respectively, etc.

Figure 1: Please add the word "age" in the title of the OX axis for better clarity of Figure 1.

Line 237: 4.2 Strengths and limitations. No section 4.1. It should be section 4.1. not 4.2.

Reviewer 2 Report

The present study is a solid analysis of data regarding emotional eating during the Covid-19 lockdown in Norway. The data and analysis seems sound, and the topic is important. The manuscript is written clearly and convincingly.

Apart from the limitations already mentioned by the authors, the main weakness to the present manuscript is that it does not state clearly enough what their analysis adds to the existing literature on this topic. The authors reference the previous literature and the well-known relation between emotional distress and emotional eating, and the present study adds further validation to this literature. However, it would be good if the authors could better emphasize or articulate if the present study adds any new perspectives on the topic.

That said, the data itself seems robust, and thus in this sense the present manuscript will be a useful addition to the literature on this topic.

Reviewer 3 Report

Thank you very much for giving me the opportunity to review your paper. The side-effects of the COVID-10 pandemic and the government intervenions are highly relevant. Your paper has a high overall quality. However, reading your manuscript, I found the following issues:

  1. It is well-known that stress can lead to emotional eating. Please substantiate what your research adds to this.
  2. I’m missing comparisons with (1) pre-/after-lockdown eating behavior (i.e., “normal situations”) and with (2) eating behavior caused by other kinds of stress, such as work stress, relationship stress, stress caused by a disease, rape, etc. that could be found in the literature. Without such comparisons, it is not possible to see if the COVID-10 pandemic and the government intervenions caused a higher than usual bad eating behavior.
  3. Maybe the economic worries could be further specified. Are they related to the country's economy or the financial well-being of the individuals?

Round 2

Reviewer 3 Report

Thank you very much for addressing my comments, to which you responded in a satisfactory manner. Keep up the good work and merry christmas!